# Iron limitation in *M. tuberculosis* has broad impact on central carbon metabolism

Monique E. Theriault[1], Davide Pisu[1], Kaley M. Wilburn[1], Gabrielle Lê-Bury [1], Case W. MacNamara [2], H. Michael Petrassi[2], Melissa Love [2], Jeremy M. Rock[3], Brian C. VanderVen [1] & David G. Russell [1✉]

*Mycobacterium tuberculosis* (*Mtb*), the cause of the human pulmonary disease tuberculosis (TB), contributes to approximately 1.5 million deaths every year. Prior work has established that lipids are actively catabolized by *Mtb* in vivo and fulfill major roles in *Mtb* physiology and pathogenesis. We conducted a high-throughput screen to identify inhibitors of *Mtb* survival in its host macrophage. One of the hit compounds identified in this screen, sAEL057, demonstrates highest activity on *Mtb* growth in conditions where cholesterol was the primary carbon source. Transcriptional and functional data indicate that sAEL057 limits *Mtb*'s access to iron by acting as an iron chelator. Furthermore, pharmacological and genetic inhibition of iron acquisition results in dysregulation of cholesterol catabolism, revealing a previously unappreciated linkage between these pathways. Characterization of sAEL057's mode of action argues that *Mtb*'s metabolic regulation reveals vulnerabilities in those pathways that impact central carbon metabolism.

[1] Department of Microbiology and Immunology, College of Veterinary Medicine, Cornell University, Ithaca, NY, USA. [2] California Institute for Biomedical Research (Calibr), La Jolla, CA, USA. [3] Department of Host-Pathogen Biology, The Rockefeller University, New York, NY, USA. ✉email: dgr8@cornell.edu

Mycobacterium tuberculosis (Mtb), the causative agent of the pulmonary disease tuberculosis (TB), has evolved to be a highly specialized and successful human pathogen. Mtb's success may be attributed to the bacterium's ability to adopt an intracellular lifestyle that overcomes nutrient availability and environmental stressors[1]. This phenotypic plasticity ensures Mtb's persistence even in the face of a robust immune response or prolonged antibiotic treatment[2–4]. Understanding the pathways that Mtb mobilizes to maintain its survival is of considerable importance to the development of new therapeutic strategies to eradicate TB.

Iron is an essential micronutrient for almost all living organisms and has been implicated in the survival of many different human pathogens[5]. The utility of iron derives from the metal's ability to transition between oxidation states, making it an important enzyme cofactor in several essential biological processes, including DNA replication[6] and electron transport[7]. Due to restricted iron availability within the host, many pathogens have evolved elaborate mechanisms for scavenging iron from host storage proteins. In most bacteria, this is accomplished with siderophores, which are small molecules with high avidity for iron[8]. Mtb possesses two such siderophores—mycobactin, a nonsoluble siderophore that remains cell wall-associated, and carboxymycobactin, the soluble form of mycobactin that is secreted into the extracellular environment[9]. Both of these siderophores chelate and bind to ferric iron ($Fe^{3+}$), which is then transported back into the bacterial cell. The iron is reduced to ferrous iron ($Fe^{2+}$), which facilitates its release for utilization or storage. Intracellular iron levels are tightly regulated by the iron-sensing transcriptional activator/repressor IdeR, as both low and high iron concentrations can be toxic to the bacteria[10].

A recent assessment of the relative fitness of Mtb in a murine challenge model has revealed that the bacteria experience different host-dependent stressors that are linked to both the physiological states and the ontogenic origin of their host macrophage populations[11–13]. In broad terms, the resident alveolar macrophages (AMs) are more permissive to bacterial growth than the recruited, blood monocyte-derived interstitial macrophages (IMs). Dual RNA-seq analysis of host and bacteria demonstrated that Mtb in IMs experienced iron deprivation as a consequence of the iron-sequestration program mobilized by their host cells[12]. These bacilli exhibit upregulation of genes involved in mycobactin synthesis (mbtA-L) and iron import (irtA-B), which contrasts directly with those bacteria residing in AMs that show upregulation of transcripts for the iron storage protein (bfrB), indicating iron-replete conditions. These observations suggest that iron availability may be one of the major factors at the immune-metabolic interface involved in the control of Mtb infection[11,14]. These data also agree with earlier evidence that the intravacuolar iron levels in Mtb-infected macrophages varied directly with bacterial fitness and inversely with macrophage activation status[15].

In addition to Mtb's requirement for iron, the bacterium is also reliant on the acquisition of nutrients to fuel its growth. There is a wealth of data demonstrating that intracellular Mtb actively catabolizes lipid substrates[16–20] (cholesterol and fatty acids) and that acquisition of these lipids is necessary for survival in the macrophage[19]. While some of the pathways involved in the regulation of lipid metabolism in Mtb have been identified[16,21,22], we are still struggling to fully understand the regulatory wiring that controls Mtb metabolism. The complexity of Mtb's metabolic program has been further revealed through recent findings demonstrating Mtb's capacity to catabolize host-derived glycolytic intermediates such as pyruvate and lactate[23,24].

During a recent high-throughput screen for compounds that exhibited enhanced activity against intracellular Mtb, we identified a compound, sAEL057, which had enhanced activity against intracellular Mtb in comparison to rich broth. Mode of action (MOA) analyses indicated that the compound dysregulates iron acquisition through the chelation of Fe. While synthetic iron chelators are approved for clinical use to prevent iron-induced disease progression and to control inflammation[25,26], the long-term administration of anti-TB drugs required for effective treatment likely limits the usefulness of such compounds for TB treatment. Nonetheless, sAEL057 has proven to be a useful tool in revealing that iron plays a hitherto unappreciated role in the assimilation of cholesterol. Our data indicate that dysregulation of iron homeostasis caused by sAEL057 differentially impacted Mtb growth on specific carbon sources, most notably cholesterol. This finding builds on an extensive body of work stressing the importance of cholesterol utilization for Mtb survival in vivo and demonstrates that both survival and cholesterol utilization can be dysregulated through iron limitation.

## Results

### sAEL057 was identified in a high-throughput screen against intracellular Mtb. In collaboration with the California Institute of Biomedical Research (Calibr, a subsidiary of Scripps), we conducted a large-scale chemical screen to identify new antimicrobials active against Mtb. The primary screen was conducted using a fluorescent readout on J774 macrophage-like cells infected with mCherry-expressing Mtb to assess intracellular drug activity, as detailed previously[27]. In addition, with the goal of trying to identify inhibitors of cholesterol metabolism, secondary screens were performed on Mtb grown in a rich broth, minimal broth supplemented with 100 μM cholesterol, and minimal broth supplemented with 100 μM palmitate.

Compounds that showed a $\log_{10}$-fold reduction in Mtb fluorescence, compared to negative DMSO controls, were denoted as a hit for each category. Of the ~1.1 million compounds screened, sAEL057 (Supplemental Fig. 1) was found to inhibit Mtb growth both in J774 macrophages and cholesterol-containing media. However, its activity did not reach the threshold to qualify as a hit in rich broth or palmitate-containing media, leading to its initial designation as a "cholesterol-dependent compound."

### Efficacy of sAEL057 is related to the metabolic status of bacteria. Although Mtb is regarded as a metabolically plastic organism capable of co-metabolizing simple carbon substrates[28], we noted previously that chemical inhibition of cholesterol metabolism could not be rescued by the addition of glucose to the cholesterol medium, whilst it could be rescued with acetate[27]. This suggests that there is a level of genetic control or cross-talk between metabolic pathways in Mtb resulting in an unusual form of carbon catabolite repression which is best revealed by chemical inhibition. To evaluate the impact of carbon source identity on sAEL057 activity, Mtb was grown in minimal media (7H9) supplemented with different carbon sources (cholesterol, oleate, acetate, glucose or OADC). Drug activity was assessed by the addition of a redox-sensitive, cell-permeable viability reagent, alamar blue, and data was analyzed across a range of dosages to determine $EC_{50}$s of sAEL057 in each media.

Consistent with the primary assay, sAEL057 treatment demonstrated activity at low micromolar levels in inhibiting Mtb growth in a carbon source-dependent fashion (Fig. 1a, b). As indicated by $EC_{50}$ values, its activity was greater when Mtb was grown in the presence of cholesterol or glucose as its sole carbon source, while activity in acetate was intermediary (Fig. 1b). sAEL057 showed reduced activity against bacterial growth in the presence of a long-chain fatty acid (OADC or oleate). In addition, when the cholesterol media was supplemented with oleate, we

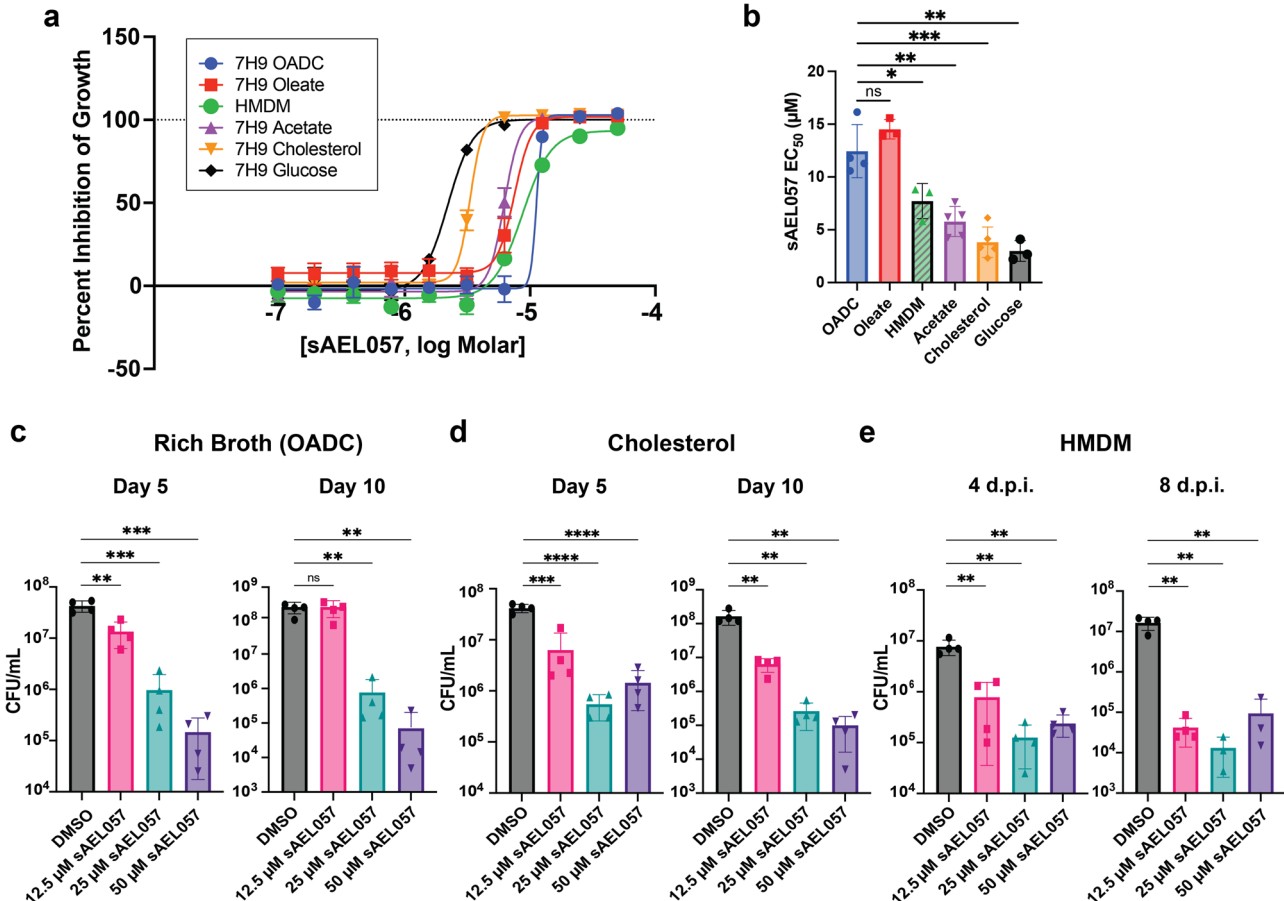

**Fig. 1 Growth inhibition of *Mtb* upon treatment with sAEL057 in various conditions. a, b** *Mtb* were grown in broth (7H9 + OADC, 7H9 + Oleate, 7H9 + Acetate, 7H9 + Glucose, and 7H9 + Cholesterol) and treated with sAEL057 at 50 μM down to 0.097 μM in a dose curve-dependent manner. Alamar blue was added on day 9 and results were measured via fluorescence on day 10. HMDMs were infected at MOI 2:1 with mCherry-expressing *Mtb* and fluorescence values were measured at 7dpi. **a** Representative graph of percent inhibition curves. Percent inhibition was determined relative to DMSO (0% inhibition) and 20 μM RIF (100% inhibition) controls. $n = 2$ technical replicates from a representative experiment. **b** EC$_{50}$ calculated using nonlinear regression analyses (log inhibitor vs response) of percent inhibition curves. $n = 3$–5 replicate experiments **c, d** *Mtb* was grown in 7H9 + OADC (**c**) or 7H9 + cholesterol (**d**), and replicate plates were used for plating CFUs on day 5 and day 10 posttreatment. $n = 4$ from two replicate experiments. **e** *Mtb*-infected HMDMs were lysed via 0.01% SDS in water at 4 dpi and 8 dpi to plate for CFU. $n = 4$ from two replicate experiments. Statistical significance and *P* values were assessed using a student's unpaired *t*-test. Error bars indicate standard deviation. Source data and *P* values for all main figures are available in Supplementary Data 1.

saw the partial rescue of the growth inhibition phenotype (Supplemental Fig. 2). CFU (colony-forming unit) counts following treatment of *Mtb* with sAEL057 indicated that the growth inhibition was a consequence of bactericidal activity in rich broth (Fig. 1c) and cholesterol media (Fig. 1d).

Finally, to assess drug activity in primary macrophages, we infected human monocyte-derived macrophages (HMDMs) at a multiplicity of infection (MOI) of 2:1 and treated them with sAEL057. The EC$_{50}$ in HMDMs was 7 μM (Fig. 1b) and CFU counts indicated strong bactericidal activity (Fig. 1e). Taken together, these results demonstrate that sAEL057, at low micromolar concentration, is able to inhibit *Mtb* growth in broth, cell-lines, and primary macrophage cultures. Intriguingly, the growth inhibition assays performed in different carbon sources suggested that cholesterol breakdown might not be the target of sAEL057 activity as the compound impacted *Mtb* growth in different carbon source media.

**Transcriptional profiling indicates sAEL057 is dysregulating iron homeostasis and cholesterol metabolism in *Mtb*.** To identify the mode of action of sAEL057 inhibition of *Mtb* growth, we performed two sets of RNA-seq experiments. Because

sAEL057 strongly inhibits *Mtb* growth in the presence of cholesterol, the first set of RNA-seq experiments were conducted on *Mtb* grown in cholesterol-supplemented media. Additionally, due to the observed bactericidal activity inside of HMDMs, we also performed transcriptional analyses on *Mtb* from HMDMs infected with mCherry-expressing *Mtb*. The bacteria were grown under both conditions (cholesterol media or intramacrophage) for two days prior to the addition of the drug or DMSO as the negative control. This ensured the bacteria had established those early transcriptional changes necessary to adapt to that specific environment *prior* to drug exposure so subsequent changes would be drug-specific.

Principal component analysis (PCA) of the transcriptional profiles across different drug treatments shows that *Mtb* response to sAEL057 is markedly divergent from other compounds, indicating that sAEL057 had a different MOA (Fig. 2a and Supplemental Fig. 3). We included the frontline drugs isoniazid (INH) and ethambutol (EMB), as well as another compound, mCLB073, an analog of previously described cholesterol inhibitors[27,29], as comparators. PCA indicated that sAEL057 induced the largest change relative to DMSO controls (Fig. 2a and

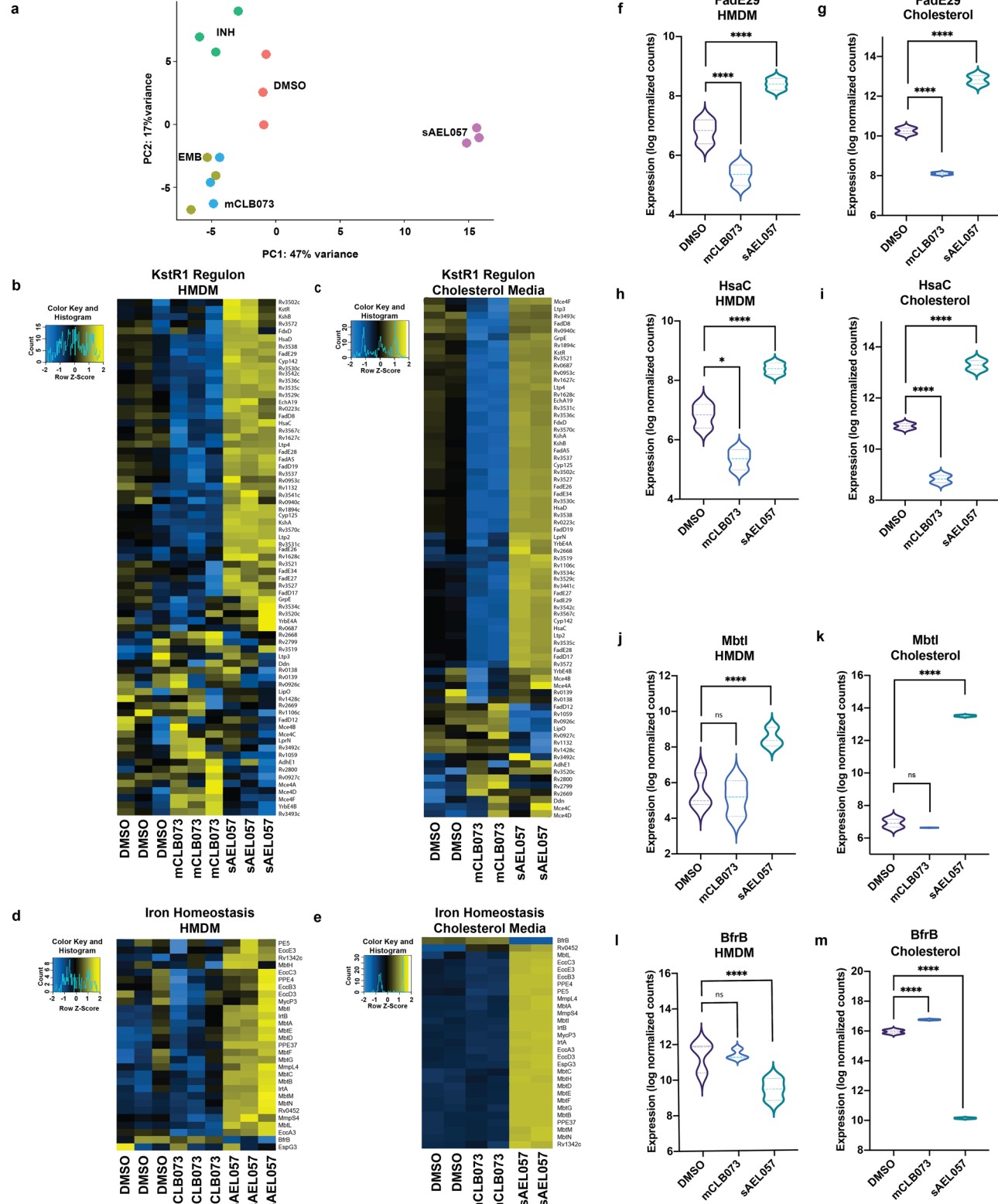

**Fig. 2 sAELO57 treatment leads to dysregulation of genes involved in iron homeostasis and cholesterol metabolism in *Mtb*. a** PCA analysis of *Mtb* transcriptomes from infected HMDMs. **b**, **c** Heatmaps showing relative expression of genes in the KstR1 regulon. **d**, **e** Heatmaps showing relative expression of genes involved with iron homeostasis. **a**, **b**, **d**, **f**, **h**, **j**, **l** HMDMs were infected at MOI 2:1 for 2 days with mCherry-expressing *Mtb* prior to administration of sAELO57 at 10 μM. Infected macrophages were sorted for mCherry positivity 2 days after sAELO57 treatment was commenced, at 4 dpi. Samples were sorted into triazole prior to RNA extraction. **c**, **d**, **g**, **i**, **k**, **m** *Mtb* was pre-grown in cholesterol-supplemented media (7H9 + cholesterol) for 2 days prior to the addition of sAELO57 at 10x MIC (23 μM). Samples were collected at 4 h posttreatment for RNA extraction. Normalized counts were used for the generation of all heatmaps. **f–m** Violin plots showing expression (in log normalized counts) of genes encoding for HsaC, FadE29, BfrB, and MbtI in the two environmental conditions. Statistical significance values based on adjusted *P* values ($p_{adj}$) in Supplementary Data 1 for respective genes.

Supplemental Fig. 3). mCLB073, belongs to a group of compounds known to activate adenylate cyclase in *Mtb* encoded by the gene *rv1625c*, and this leads to inhibition of cholesterol catabolism[27,29]. Intriguingly, PCA analysis showed a clear separation between sAEL057- and mCLB073-treated *Mtb* despite both compounds showing the highest activity in the presence of cholesterol, suggesting that the compounds differ in their MOA (Fig. 2a and Supplemental Fig. 3). Indeed, when we examined those genes involved in cholesterol catabolism, sAEL057 was associated with upregulated expression of many Kstr1 regulon genes, while mCLB073 induced their downregulation (Fig. 2b, c, f–i). KstR1 is a transcriptional regulator which is de-repressed by the cholesterol catabolite, 3OCh-CoA, an early intermediate in the degradation of cholesterol[30]. The KstR1 regulon includes enzymes necessary for catabolism of the cholesterol sidechain and A and B rings[31]. Interestingly, *Mtb* treated with sAEL057 showed no impact on cholesterol import or the KstR2 regulon expression (Supplemental Fig. 4). KstR2 is another transcriptional repressor which is de-repressed by the cholesterol catabolite HIP-CoA[30]. This leads to the expression of enzymes required for the degradation of cholesterol's C and D rings[32]. These data suggest that sAEL057-treated *Mtb* are not inhibited in import or initial degradation of cholesterol, which implies that this compound does not act through inhibition of early cholesterol breakdown.

The largest transcriptional perturbation seen upon sAEL057 treatment was in those genes involved in iron metabolism (Fig. 2d, e, j–m). In both HMDMs and cholesterol media, *Mtb* treated with sAEL057 significantly upregulated mycobactin synthesis (*mbtA-L*) and iron import (*irtA* and *irtB*) genes and significantly downregulated the iron storage protein gene *bfrB*. This shift was especially marked in cholesterol media where we observed a 6-$\log_{10}$ fold decrease in *bfrB* expression (Fig. 2m) and a 7-$\log_{10}$ fold increase in *mbtI* expression (Fig. 2k). In addition, in cholesterol media, sAEL057 treatment also leads to upregulation of genes involved in the ESX-3 secretion system (Fig. 2e), which is known to be required for iron import and homeostasis[33]. This transcriptional signature is consistent with an iron deprivation response, and interestingly, overlaps almost entirely with the iron-related transcriptional response that was detailed in *Mtb* in blood monocyte-derived IMs isolated from infected mouse lungs analyzed by Dual RNA-seq[14]. The IM host cell population is a markedly better controller of *Mtb* growth, and iron deprivation appears to be an important component of this control.

One of our standard steps in MOA studies is the isolation of spontaneous resistance mutants under selection from compounds of interest. Attempts to isolate resistance mutants against sAEL057 were unsuccessful, which combined with the RNA-seq data, suggested sAEL057 may not have a specific genetic target but could be blocking *Mtb*'s capacity to acquire or utilize iron.

**Gallium synergizes with sAEL057 to inhibit *Mtb* growth in cholesterol media but not in rich broth**. Due to the striking iron deprivation signature revealed by the RNA-seq analyses, we chose to examine how the addition of gallium nitrate to the growth media impacted drug efficacy. Gallium, a post-transition metal, has a similar ionic radius to iron and competes for protein binding, however, under physiological conditions it is not able to be reduced and thus renders iron-dependent enzymes nonfunctional. Gallium has been shown to disrupt many iron-dependent biological processes[34] and in *Mtb*, in particular, has been used as a bactericidal compound to disrupt iron metabolism in both extracellular and intramacrophage *Mtb*[35,36].

First, we tested whether gallium had a synergistic effect on sAEL057 treatment. The impact of gallium on *Mtb* growth in cholesterol media was tested at three different concentrations

(Fig. 3a–c) and we observed that the addition of gallium resulted in a sharp decrease in the $EC_{50}$ of sAEL057 (Fig. 3b), indicating a synergistic impact with sAEL057. In line with previous studies[35,36], the addition of gallium alone came at a fitness cost to *Mtb* (Fig. 3c). A similar trend was observed when the same set of experiments were performed on *Mtb* grown in glucose media, however, the degree of synergy was reduced compared to cholesterol media (Supplemental Fig. 5). Lastly, there was little impact on sAEL057's activity with gallium addition on *Mtb* growth in rich broth conditions (Fig. 3d, e). Gallium alone does not confer fitness cost to *Mtb* in this environment (Fig. 3f). These results indicate that *Mtb* has a greater dependency on iron when reliant on cholesterol as a limiting carbon source, as compared to growth in nutrient-rich conditions. A prior study, utilizing an in vitro model for iron deprivation, noted that cholesterol utilization genes were upregulated upon iron starvation in *Mtb*[37]. Our transcriptional data are consistent with that finding, and these experiments provide functional evidence linking iron homeostasis and cholesterol metabolism.

**Mass spectrometry analyses reveal that sAEL057 binds and forms a complex with ferrous iron**. The RNA-seq data showed that sAEL057 treatment causes dysregulation of iron homeostasis in *Mtb* and results from our functional studies indicated that this dysregulation may be a consequence of iron chelation. We, therefore, performed liquid chromatography-tandem mass spectrometry (LC-MS/MS) to assess whether sAEL057 binds to iron. We analyzed sAEL057 in the presence and absence of ferrous ($FeCl_2$) and ferric ($FeCl_3$) iron. LC-MS/MS demonstrated a shift in the sAEL057 peak when ferrous iron was present (Supplemental Fig. 6). Furthermore, based on the known masses of sAEL057 and $Fe^{2+}$, a calculated monoisotopic mass predicts that three sAEL057 molecules bind to a single $Fe^{2+}$ (Supplemental Fig. 6d) Fe can form six-coordinate covalent bonds which are predicted to be made via electron donation from the nitrogen atoms of sAEL057.

**sAEL057 alters cholesterol metabolism in *Mtb* downstream of early ring degradation**. The transcriptional data indicated that sAEL057 treatment led to upregulation of the KstR1 regulon, which includes cholesterol sidechain and ring degradation genes (Fig. 2). To test whether *Mtb* treated with sAEL057 was capable of enzymatic breakdown of cholesterol, we utilized [4-$^{14}$C]-cholesterol (Fig. 4a) to serve as a direct measure of A ring degradation[30,38]. In this assay, when the A-ring of [4-$^{14}$C]-cholesterol is broken down by the bacteria, it is released as $^{14}CO_2$, and we have used this assay previously as a relative measure of cholesterol catabolism[18]. The amount of $^{14}CO_2$ released during sAEL057 treatment was comparable to DMSO treated controls (Fig. 4b). Although cholesterol degradation is a complex process, this result is broadly consistent with the conclusion that sAEL057-treated bacteria are not deficient in the early stages of cholesterol catabolism.

To assess whether sAEL057 exposure altered central metabolism downstream of cholesterol ring degradation, we used the bacterial reporter strain *prpD'*::GFP[39]. PrpD (Rv1130) is one of the enzymes involved in the methyl citrate cycle (MCC), a key enzymatic process in *Mtb* for converting the toxic cholesterol breakdown product propionyl-CoA into methyl citrate for incorporation into the TCA cycle[21,40]. The expression of *prpD* is positively regulated by propionyl-CoA accumulation during cholesterol and propionate metabolism specifically[18,39,41,42]. Therefore, reduced *prpD* expression, as quantified by GFP fluorescence in the reporter strain, would indicate a reduced accumulation of cholesterol-derived propionyl-CoA. Using the *prpD'*::GFP reporter strain and the gating strategy in Supplemental Fig. 7, we noticed that there was a concentration-dependent impact on propionyl-CoA accumulation with sAEL057 treatment (Fig. 4c–d). Both the percent of GFP-positive bacteria

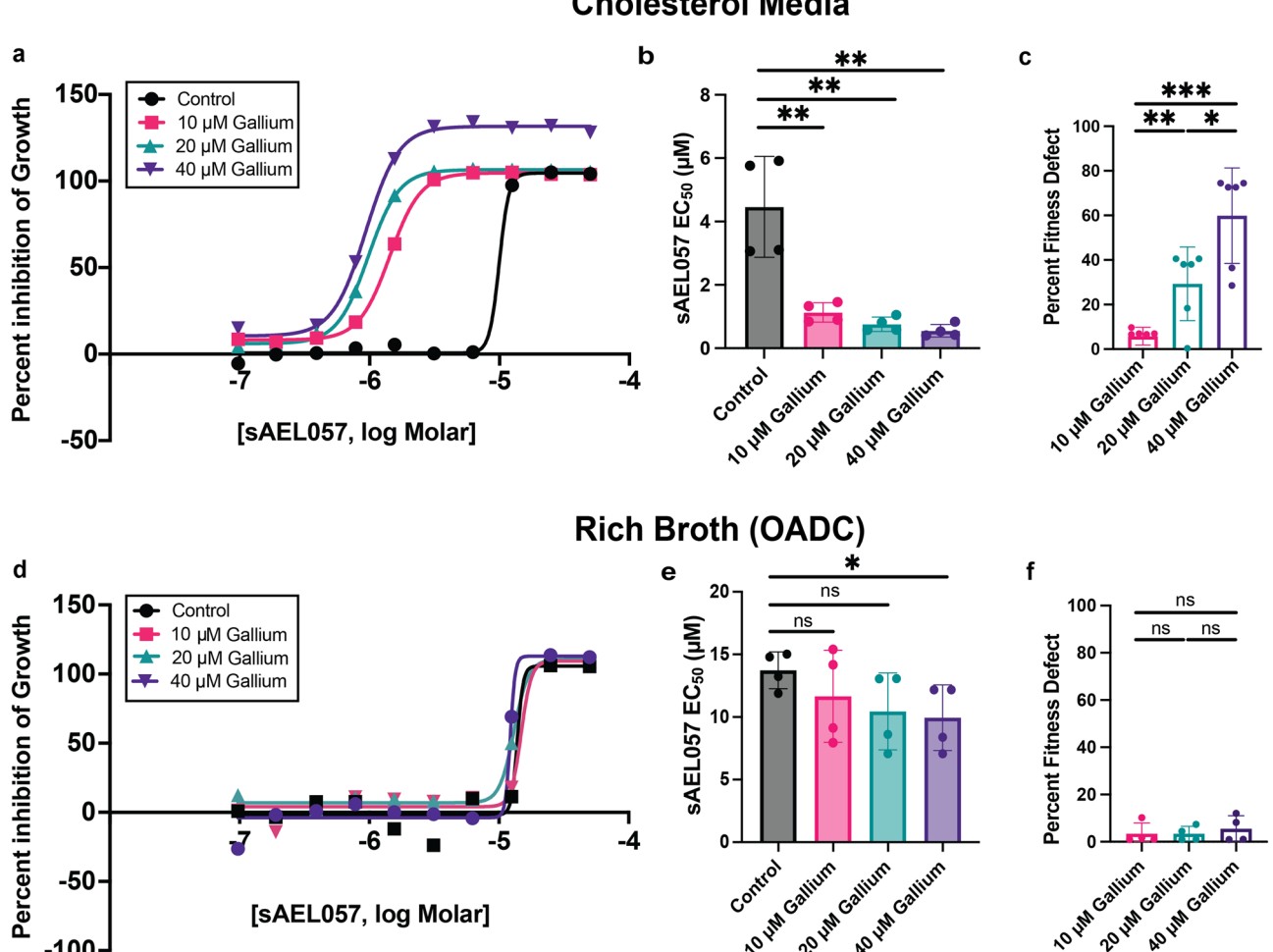

**Fig. 3 Gallium addition synergizes with sAEL057 treatment to inhibit *Mtb* growth in cholesterol media but not in rich broth.** *Mtb* were grown in cholesterol-supplemented media (**a**–**c**) or rich broth (**d**–**f**) and treated with sAEL057 at 50 µM down to 0.097 µM in a dose curve-dependent manner. Alamar blue was added at day 9 and results were measured via fluorescence on day 10. **a**, **d** Representative graphs of percent inhibition curves. Percent inhibition was determined relative to DMSO (0% inhibition) and 20 µM RIF (100% inhibition) controls. $n = 2$ technical replicates from a representative experiment. **b**, **e** EC$_{50}$ values calculated using nonlinear regression analyses (log inhibitor vs response) of percent inhibition curves. $n = 4$ from two replicate experiments. **c**, **f** % Fitness defect was calculated by dividing gallium only (without sAEL057) controls by DMSO controls and multiplying by 100 in the given media condition. $n = 4$–6 from two replicate experiments. Statistical significance was assessed using a student's unpaired *t*-test. Error bars indicate standard deviation. Source data and *P* values for all main figures are available in Supplementary Data 1.

(Fig. 4c) and the median GFP fluorescence of GFP-positive bacteria (Fig. 4d) were reduced upon sAEL057 treatment. This effect was detectable 4 h after drug treatment and was sustained over time, with 48 h showing the most marked impact on propionyl-CoA induced *prpD* expression. The combined results of the *prpD*::GFP reporter and $^{14}CO_2$ release assay indicate that sAEL057 is preventing full assimilation of cholesterol, possibly at a step after early ring degradation but prior to the MCC. The cholesterol degradation pathway contains enzymes such as HsaC, an extradiol dioxygenase that is known to be iron-dependent[20]. This could account for the reduction in complete cholesterol breakdown and the upregulation of KstR1 regulon genes as a compensatory response to reduced metabolic output.

**Knockdown of *Mtb*'s iron-sensing transcription factor, IdeR, leads to a larger growth defect in cholesterol media compared to a rich broth.** As an independent confirmation that iron dysregulation results in a growth deficient phenotype under nutrient-specific conditions, we generated a Tet-inducible *ideR* knockdown

strain using the mycobacterial CRISPRi system developed by ref. [43]. IdeR is an iron-response regulator that, under iron-limiting conditions, up-regulates those genes responsible for iron acquisition in vivo[44]. Upon the addition of ATc (anhydrous tetracycline), we observed an early and sustained downregulation of *ideR* expression as assessed by RT-qPCR (Supplemental Fig. 8). There was no effect on the growth of the *ideR* knockdown strain in rich broth unless sAEL057 was also present (Fig. 5a and Supplemental Fig. 9). However, when we induced the *ideR* knockdown in *Mtb* grown in cholesterol media, the bacteria exhibited a reduced growth rate that was reduced further by the presence of sAEL057 (Fig. 5b and Supplemental Fig. 9). Thus, not only did we observe a synergistic growth defect when *ideR* knockdown was combined with sAEL057 treatment, the link between iron regulation and cholesterol-dependent sAEL057 activity was phenocopied in the *ideR* knockdown strain. This trend was similar to *Mtb* growth in glucose media (Supplementary Figs. 9, 10), however, the impact of *ideR* knockdown was not as marked as seen in cholesterol media. These results support the conclusion that there is a linkage between iron homeostasis and

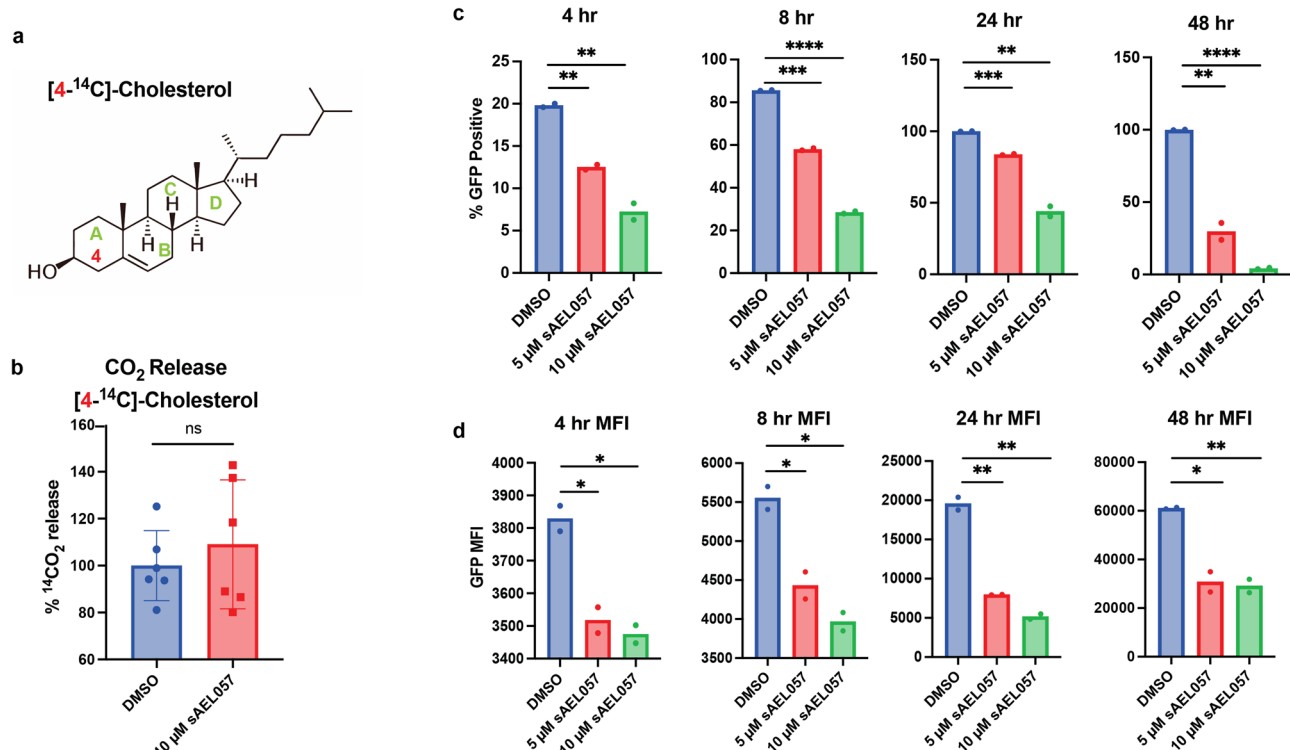

**Fig. 4 sAEL057 treatment leads to altered cholesterol metabolism in *Mtb*. a**, **b** *Mtb* was pre-grown for 6 days in cholesterol-supplemented media prior to the addition of sAEL057 at 10 μM or DMSO control. [4-$^{14}$C] radiolabeled cholesterol was added at 24 h posttreatment and $^{14}CO_2$ was measured 6 h later. $n = 6$ from two replicate experiments. **c**, **d** The bacterial reporter, *prpD'*::GFP *smyc'*::mCherry, was used to assess levels of propionyl-CoA with sAEL057 treatment in cholesterol-supplemented media. **c** % GFP-positive bacteria were determined by gating on mCherry positive bacteria and measuring the GFP-positive signal using a BD Symphony analyzer. **d** Bacteria positive for both mCherry and GFP fluorescence were analyzed to determine the GFP median fluorescence intensity (MFI). $n = 2$ from replicate experiments. Statistical significance was assessed using a student's unpaired *t*-test. Error bars indicate standard deviation. Gating strategy displayed in Supplemental Fig. 7. Source data and *P* values for all main figures available in Supplementary Data 1.

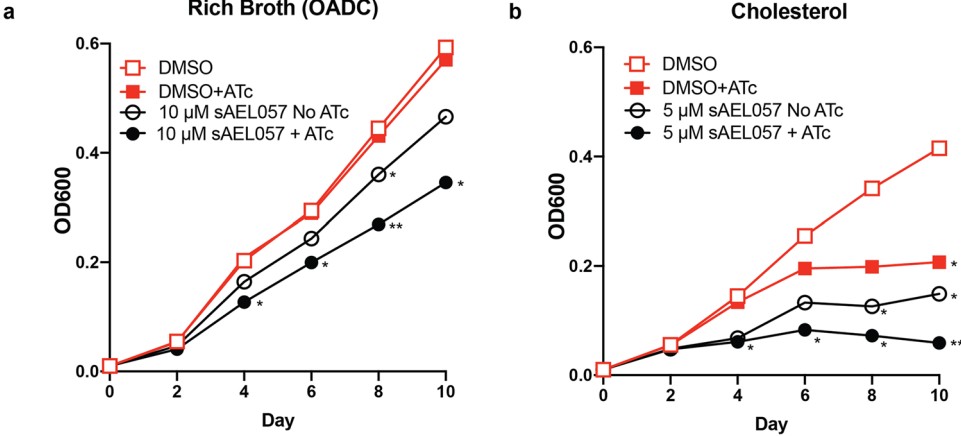

**Fig. 5 Growth phenotype with knockdown of *ideR* is more pronounced in cholesterol vs rich broth conditions.** IdeR knockdown strain was generated in Erdman *Mtb* using a CRISPRi construct. The knockdown strain was grown in media supplemented with OADC (**a**) or cholesterol (**b**) and OD$_{600}$ values were measured every 2 days. +ATc = induced *ideR* knockdown, no ATc = uninduced controls. Red lines: DMSO controls (no sAEL057). Black lines: sAEL057 treated at 10 μM in OADC (**a**) or 5 μM in cholesterol media (**b**). Statistical significance was assessed using a student's unpaired *t*-test. Source data and *P* values for all main figures are available in Supplementary Data 1.

carbon utilization in *Mtb*, and that the bacteria exhibit a heightened sensitivity to iron starvation when growing in the presence of cholesterol.

## Discussion

Chemotherapeutic treatment of tuberculosis remains protracted and frequently results in failure or in the selection of drug-resistant mutants. We, therefore, require both new drugs and information on new drug targets that are revealed in the in vivo environment. Previously we conducted a large-scale chemical screen for compounds active against *Mtb* within the host macrophage[27]. Results from this screen, and from our ongoing transcriptional and metabolic profiling of both macrophage and *Mtb* during in vivo infection[11–13,45] have led us to appreciate that

the in vivo environment in the different macrophage host lineages plays a major role in shaping bacterial metabolism. *Mycobacterium* spp. exhibit fairly extensive genome downsizing, with *M. leprae* the most extreme example with fewer than 2000 functional genes[46] compared to *Mtb* with around 4300 genes[47]. That contrasts with *M. smegmatis* with 6500 genes[48]. Many studies have commented on these gene numbers and linked this to the loss of specific functions[49]. However, we believe of equal consequence is how gene loss will have impacted the regulation of metabolism. Because *Mtb* does not experience or need to respond to the diverse environmental conditions encountered by *M. smegmatis*, it is reasonable to speculate that *Mtb* has evolved to link physiological and metabolic pathways to environmental cues relevant to intracellular infection. This may indicate why chemical inhibition of cholesterol degradation can be rescued by acetate but cannot be rescued by glucose[27], as the consequences of an unusual form of carbon catabolite repression. sAEL057 was identified as active against intracellular bacteria and shows enhanced activity in the presence of cholesterol. However, phenotypic and functional characterization of sAEL057 revealed it is an iron chelator and indicates a previously unappreciated linkage between iron homeostasis and cholesterol metabolism. A graphic model of this proposed interplay is shown in Fig. 6.

This is an interesting association because it coalesces an extensive body of literature dating back to the 1970s when Kochan first coined the term "nutritional immunity"[50]. The original premise that hosts limit pathogen growth by immune-mediated iron-sequestration mechanisms has been proven true in many infections. Subsequent gene mapping studies identified NRAMP1 (natural resistance-associated macrophage protein 1), which was found to localize to phagosomal membranes and pump $Fe^{2+}$ and $Mn^{2+}$ out of the phagosome[51,52]. To overcome the reduced iron availability inside the phagosomal environment in which it primarily resides, *Mtb* relies on the ability of its siderophores to strip iron from host storage proteins (transferrin, etc.) and heme molecules[53]. Our data indicate that dysregulation of iron homeostasis through pharmacological and genetic intervention reduces intracellular survival. Both the macrophage and the *Mtb* bacterium internalize iron in its ferric form, through its association with transferrin and mycobactin respectively. Mass spec. analysis indicates that sAEL057 preferentially binds iron in its ferrous form, which implies that the drug is able to enter the bacterium and sequester cytosolic $Fe^{2+}$. Similar antibacterial activities have been reported for the ferrous iron chelator 2,2-Bipyridyl in *Acetinobacter baumanni* through a mechanism also synergistic with gallium[54].

Several transcriptional and genetic studies have highlighted the essentiality of lipid metabolism for *Mtb* intracellular survival[17,19,22,38,45,55–58] and one previous study noted that the expression of genes required for cholesterol utilization were upregulated in iron-deficient *Mtb*[37]. We propose that it is the perturbation of iron homeostasis through iron chelation by sAEL057 that leads to a defect in cholesterol metabolism. The iron deprivation signature is not seen in *Mtb* treated with an inhibitor of cholesterol catabolism, mCLB073. Thus, iron appears to be the critical factor in the association between these two metabolic programs. We have not yet identified which step(s) of cholesterol utilization are affected by iron deprivation, and this would be an interesting point to address in future studies. We hypothesize that the reduced metabolic plasticity shown by *Mtb* may be a consequence of genome downsizing as *Mtb* evolved from a saprophytic, soil-dwelling organism to a specialized, intracellular human pathogen.

Our prior data from Dual RNA-seq and scRNA-seq analyses highlight the importance of iron-sequestration and the ability to control *Mtb* infection in vivo[13,14]. We, therefore, believe that

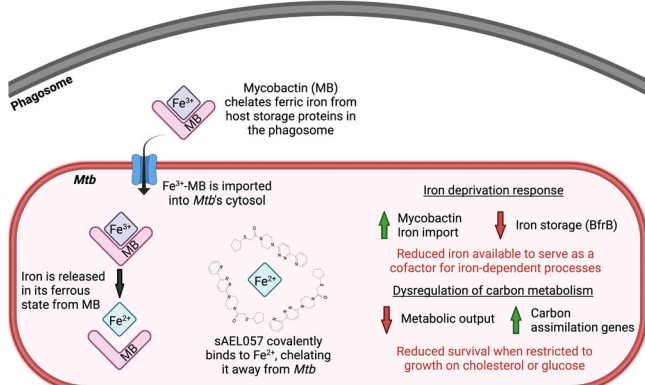

**Fig. 6 Proposed model for sAEL057 mode of action.** Mycobactin (MB) chelates ferric iron ($Fe^{3+}$) from host storage proteins inside the phagosome. Iron-laden mycobactin is imported into the bacterial cytosol through the IrtA/B transporter. Ferric iron is reduced to ferrous iron ($Fe^{2+}$) to facilitate release from mycobactin. Nitrogen atoms of sAEL057 molecules form coordinate covalent bonds with $Fe^{2+}$ through the donation of electrons to $Fe^{2+}$. The binding of sAEL057 to iron makes it inaccessible to *Mtb*, resulting in a reduced availability of intracellular iron. This leads to an iron deprivation transcriptional response, driven by the iron-sensing transcription factor IdeR, and is demonstrated by the upregulation of mycobactin and iron import genes and downregulation of iron storage genes. Additionally, we predict that catabolism of certain carbon substrates (such as cholesterol and glucose) have an increased reliance on iron-dependent proteins as compared to catabolism of other carbon substrates (such as long-chain fatty acids). Therefore, a secondary consequence of sAEL057's iron chelation is altered cholesterol catabolism leading to reduced metabolic output and a compensatory upregulation of genes involved in the early steps of the enzymatic breakdown of cholesterol. Ultimately, this results in reduced growth and survival when *Mtb* is restricted to specific single carbon substrates. Created with BioRender.com, publishing license in Supplementary Material.

targeting bacterial pathways for iron acquisition would enhance the antimicrobial activities of the host at restricting *Mtb* survival during infection. Here we show that restricting iron availability will also impact cholesterol utilization, another pathway of central importance to intracellular survival. The mechanisms linking these two processes will require additional analysis, however, the coupling of essential metabolic processes represents a possible "weakness" in *Mtb* that is potentially druggable. Lastly, this work highlights the power of "chemical genomics", through the combination of small molecule inhibitors and genetic manipulation.

## Materials and methods

**Bacterial strains and preparation for infection.** Bacterial strains were inoculated into roller bottles and grown for 3 days to mid-log prior to infection. CDC1551 *smyc'*::mCherry constitutive reporter strain was used for initial high-throughput screening. Validation experiments were conducted using the Erdman P606'::mKO-tetON inducible reporter strain. Erdman *smyc'*::mCherry was used for Dual RNA-Seq experiments. Erdman wildtype (lacking a fluorescent reporter gene) was used for all experiments where CFUs or alamar blue was used as a readout. At the time of infection, bacteria were pelleted, resuspended in basal uptake buffer (BUB—25 mM dextrose, 0.5% bovine serum albumin, 0.1% gelatin, 1 mM $CaCl_2$, 0.5 mM $MgCl_2$ in PBS), passed through a tuberculin syringe 10 times, and resuspended in macrophage media.

**Phenotypic high-throughput screening to identify potent compounds.** J774 cells were obtained from ATCC. Cells were maintained in DMEM + 10% FBS + 2 mM L-glutamine + 1 mM sodium pyruvate (complete DMEM). J774 cells were seeded in 384-well optically clear, black plates with ~30,000 cells/well 24 h prior to infection. J774 cells were then infected with CDC1551 *smyc'*::mCherry at an MOI of 10:1 (~330,000 bacteria/well). Assay plates were stored in a 37 °C, 5% $CO_2$ incubator for 1 h prior to compound treatment. Compounds were added at a final

concentration of 5 μM. DMSO and 10 μM Rifampicin were used in all assays as negative and positive controls, respectively. Assay plates were incubated for 7 days prior to analysis by a fluorescence plate reader. The compound activity was analyzed by mCherry fluorescence index relative to DMSO controls. Additional screens were conducted on *Mtb* growth in the following conditions: 7H9 + OADC, 7H9 + 100 μM palmitate, or 7H9 + 100 μM cholesterol. Compounds were added at a final concentration of 5 μM. DMSO and 10 μM Rifampicin were used in all assays as negative and positive controls, respectively. Alamar blue was added on day 9, and fluorescence values were assessed on day 10 using a plate reader. Compounds which demonstrated a 1 log reduction in alamar blue fluorescence were binned as a hit in that screening environment.

**Dual RNA-sequencing of infected human monocyte-derived macrophages.** Human monocyte-derived macrophages (HMDMs) were infected with Erdman *smyc*'::mCherry reporter *Mtb* strain at MOI 2:1. At 2 dpi, experimental compounds were administered at 10 μM, INH at 67.5 ng/mL, and EMB at 5 μg/mL. Infected HMDMs were treated for 48 h prior to live sorting using a Biorad S3 sorter. 40,000 mCherry positive cells were sorted into 700 μL of Trizol. To remove host RNA, Trizol samples were centrifuged and ~75% (~500 μL) of Trizol supernatant was transferred to a new tube. An additional 400 μL of Trizol and 150 μL of zirconia beads were added to the tubes containing the bacterial pellet and 25% of the remaining Trizol. These tubes were placed on a beat beater for two cycles of 1 min each, with 2 min of rest on ice in between. After the bead beating, approximately 75% of host RNA in Trizol was added to be bacterial lysate tubes. Chloroform was added to Trizol tubes (200 μL of chloroform for 1 mL of Trizol) and tubes were mixed via shaking vigorously. Trizol:chloroform tubes were centrifuged for 15 min and the aqueous phase (~1 mL) was transferred to a fresh tube. Two microliters of glycoblue and 500 uL of isopropanol were added to precipitate host and bacterial RNA. RNA was washed in 75% cold ethanol and RNA was eluted in 12 μL of NCFW.

**Compound treatment of bacteria grown in cholesterol-supplemented broth.** Wild-type Erdman *Mtb* was grown in 7H9 + OADC for 2 days in a roller bottle. The culture was then washed and transferred to 7H9 + cholesterol (7H9 powder, casitone, MES buffer, 100 μM cholesterol, pH = 6.6) and grown for an additional 2 days. Compounds were then administered at 10x MIC for 4 h. After 4 h of treatment, the bacterial cells were pelleted and resuspended in GTC buffer to halt transcription. RNA extraction was performed as described below.

**Bacterial RNA extraction.** Bacterial pellets in GTC buffer were washed with PBS + 0.05% Tween-80. Bacterial cells were lysed via 5 mg/mL lysozyme treatment for 15 min at room temperature followed by hot Trizol bead beating for one cycle of 2 min. After bead beating, chloroform was added and the tubes were centrifuged to separate phases. The aqueous phase was then added to an equal volume of pure ethanol. RNA was then purified using a Qiagen RNeasy Kit.

**Carbon source assays.** Wild-type Erdman was pre-grown in 7H9 + OADC to mid-log. At the time of assay setup, bacteria were pelleted and washed once in carbon source media. The bacteria were then resuspended in carbon source media to an OD of 0.02. About 100 μL/well of bacteria was added to duplicate 96-well black bottom plates (assay plates). Drug plates were made by performing twofold dilutions starting at 100 μM in respective carbon source media. About 100 μL/well of the drug plate was added to the assay plate, for a final dose range of 50 μM down to 0.097 μM. Carbon source media used was: 7H9 + OADC, 7H9 + cholesterol, 7H9 + oleate, 7H9 + acetate, and 7H9 + glucose. Plates were stored at 37 °C for 9 days. On day 9, alamar blue was added at a final concentration of 0.03 mg/mL to each well. Plates were read 20–22 h after the addition of alamar blue. Percent inhibition was calculated relative to positive (20 μM RIF) and negative (DMSO) controls.

**Gallium synergy assays.** Wild-type Erdman was pre-grown in 7H9 OADC to mid-log. At the time of assay setup, bacteria were pelleted and washed twice in respective carbon source media (7H9 + cholesterol, 7H9 + OADC, or 7H9 + glucose). The bacteria were then resuspended in respective carbon source media containing 80, 40, 20, or 0 μM gallium nitrate to an OD of 0.02. About 100 μL/well of bacteria was added to duplicate 96-well black bottom plates (assay plates). Drug plates were made by performing twofold dilutions starting at 100 μM in 7H9 + cholesterol. About 100 μL/well of the drug plate was added to the assay plate, for a final dose range of 50 μM down to 0.097 μM. Final gallium concentrations in the assay plates were 40, 20, 10 μM, and no gallium control. Plates were stored at 37 °C for 9 days. On day 9, alamar blue was added at a final concentration of 0.03 mg/mL to each well. Plates were read 20–22 h after the addition of alamar blue. Percent inhibition was calculated relative to positive (20 μM RIF) and negative (DMSO) controls.

**LC-MS/MS method for analysis of sAEL057 and Fe(II) complexes with sAEL057.** The three conditions tested for LC-MS/MS analyses were: 20 μM sAEL057 in sterile water, 20 μM sAEL057 plus 10 μM $FeCl_2$ in sterile water, and 20 μM sAEL057 plus 10 μM $FeCl_3$ in sterile water. The analysis of the ligands and

Fe (II) complexes were done using Sciex X500B QTOF mass spectrometer (Sciex, Framingham, MA), coupled to an ExionLC HPLC system (Sciex) and operated in the positive ion mode. The sample injection volume was 10 μL. The column used for analysis was Luna C18 (2) 3 μm, 200 Å column (100 mm × 2 mm inside diameter (Phenomenex, Inc., Torrance, CA). Mobile phase A was 0.1% formic acid and mobile phase B was 100% acetonitrile. The flow rate was set to 200 μL/min. The gradient condition was as follows: starting solvent 2% B, held for 2 min, increased to 45% B over 5 min, and further increased to 98% B in 1 min and held at 98% B for 4 min, and finally decreased to 2% B followed by a 3 min equilibration. The MS was operated in the ESI positive ion mode, scanning from m/z 100 to 1400 in IDA mode, DP −50 V and CE 10 V for TOF MS and set with IDA criteria as a peptide, maximum candidate ion for MS2 was set as 20 with intensity threshold exceeds 100 cps having dynamic CE for MS/MS and dynamic background subtraction. The following optimized operating conditions were used: voltage of 5.5 kV, nebulizer gas and heater gas of 20 psi, curtain gas 20, collision gas 7, source temperature of 325 °C, and accumulation time of 0.15 s.

**Cholesterol breakdown assay using *prpD* reporter strains.** The reporter strain *prpD*'::GFP *smyc*'::mCherry was pre-grown in 7H9 + OADC to mid-log. Bacteria were washed 2x in 7H9 + cholesterol, before being resuspended in 7H9 + cholesterol to an OD of 0.05 in a T-25 flask. An aliquot from the cultures in 7H9 + OADC were collected. Cultures were incubated for 2 days in 7H9 + cholesterol prior to assay start. An aliquot was removed to serve as time zero. sAEL057 was added to a final concentration of 10 or 5 μM. DMSO served as a negative control. Samples were collected 4, 8, 24, and 48 h post sAEL057 or DMSO addition and fixed overnight in PFA at 4 °C. Samples were centrifuged and resuspended in PBS prior to analyses on BD Symphony.

**$^{14}CO_2$ release assays.** Wild-type Erdman *Mtb* was pre-grown in 7H9 + OADC media until it reached the mid-log phase. Then, the bacteria were moved into 7H9 + cholesterol media, matched to an $OD_{600}$ of 0.1, and grown for 6 days to the mid-log phase. DMSO or 10 μM sAEL057 was added, and 24 h later 1.0 μCi of [4-$^{14}C$]-cholesterol was added to each of the cultures in standing vented T-25 flasks. Each vented flask was sealed inside an air-tight vessel alongside an open vial containing a strip of filter paper coated in 0.5 mL of 1.0 M NaOH. The vessels were incubated for 5 h at 37 °C. Each NaOH vial was recovered, the contents were washed with 0.5 ml of 1.0 M HCl to neutralize the NaOH and recover the base soluble $Na2^{14}CO3$, and the $Na2^{14}CO3$ was quantified by scintillation counting. The radioactive counts per minute were normalized to the $OD_{600}$ of each culture, and these values were used to calculate the % $CO_2$ released relative to the DMSO control.

**Statistics and reproducibility.** Student's unpaired *t*-tests were used for most data comparisons unless otherwise noted in the figure legends. $EC_{50}$ values were calculated using nonlinear regression analyses (log inhibitor vs response) of percent inhibition curves. For each biological replicate, an $EC_{50}$ was generated from 2–3 technical replicate wells. Therefore, individual data points represent a biological replicate.

**Reporting summary.** Further information on research design is available in the Nature Research Reporting Summary linked to this article.

## Data availability

The transcriptional datasets resulting from the RNA-seq analyses are deposited on the Gene Expression Omnibus database under accession GSE196816. Source data for all main figures are included in Supplementary Data 1. All other data is available upon reasonable request to DGR.

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

## Acknowledgements

The authors would like to acknowledge the contributions of Nicole L. Kushner and Amanda C. Brown to the initial identification of sAEL057. We also wish to thank Ruchika Bhawal and other members of the Cornell Institute of Biotechnology Metabolomics and Proteomics facility for conducting and analyzing LC-MS/MS of sAEL057. Lastly, thank you to Jen Grenier and Ann Tate of the Cornell Institute of Biotechnology Genomics Core for conducting and processing RNA-seq samples. This work was supported by grants for the Bill and Melinda Gates Foundation (OPP1108452) and National Institutes of Health awards (AI134183 and AI155319) to D.G.R.

## Author contributions

M.E.T., D.P., K.M.W., and G.L.-B. performed the experiments. D.G.R., M.E.T., D.P., C.W.M., H.M.P., M.L., J.M.R., and B.C.V. contributed to the design of the experiments. M.E.T. and D.G.R. wrote the manuscript. M.E.T., D.P., K.M.W., G.L.-B., C.W.M., H.M.P., M.L., J.M.R., B.C.V., and D.G.R. edited the manuscript.

## Competing interests

The authors declare no competing interests.
