## [Peer Review File · Communications Biology]

Reviewers' comments:

Reviewer #1 (Remarks to the Author):

The authors have identified a compound sAEL057 by a high-throughput screen, which demonstrates inhibitory activity on Mtb grown in conditions with cholesterol as the primary carbon source by limiting Mtb's access to iron. The methods are well-described and manuscript reads well. While the authors have discussed the relevance of disrupting cholesterol metabolism and iron sequestration in controlling Mtb, the in vivo relevance of this study requires a more detailed explanation.

It is unclear from the title and abstract what exactly the authors are proposing with respect to sAEL057. To establish the role of this compound in iron sequestration, a direct titration assay or a competition assay would be beneficial.

Any specific reason for not using an iron chelator as a comparator in PCA to understand MOA?

A small section in discussion and a proposed model figure will be beneficial for the readers to understand the interrelationship proposed by the authors.

Line 136 mentions MOI of 1:1 for assessing drug activity in HMDMs while line 376 mentions at MOI 2:1. Kindly clarify.

Reviewer #2 (Remarks to the Author):

“Iron limitation in M. tuberculosis has broad impact on bacterial metabolism revealing alternative routes to novel therapeutics” by Theriault M.E. et al. describes a study that identifying a new inhibitor of Mycobacterium tuberculosis replication in macrophage, and it reveals that this inhibitor is a iron chelator able to inhibit the use of cholesterol by the bacteria.

This study is of great importance because it adds to the limited knowledge concerning the link between metal homeostasis and carbon metabolism, an aspect of bacterial physiology that has only recently been emphasised (PMID: 34080971). The authors posing special attention to link between cholesterol degradation and iron homeostasis.

However, the work shows some limitations as follows:

1. Unfulfilled expectations

The title is full of expectation, as the work focuses only on cholesterol, leaving out the phenotype observed on glucose. It is unclear why the effect of sAEL057 on growth in 7H12 glucose was not further investigated by the authors. It would add value to the work by better supporting the "broad impact of iron limitation on bacterial metabolism". The experiments shown in Figures 3 and 5 could be completed on Mtb grown on 7H12 glucose.

2. High throughput screening.

The minimum medium used for screening bacterial cultures is not specified in the materials and methods. In addition, a figure would make it clear how the authors identified the compound sAEL057. The authors could insert a figure like the one used in their previous work (figure 1 of reference 25).

3. 7H12 medium

It is not clear whether Rich Broth consists of 7H12 or 7H9 supplemented with OADC. And therefore, if Rich Broth consists of 7H9 supplemented with OADC, the experiments in figures 1, 3 and 5 should be repeated using as control condition 7H12 + OADC. 7H12 contains a pancreatic digest that represents a nutritional stimulus that cannot be underestimated.

4. Lines 116-118

In the sentence 'we noted [...] could not be rescued by addition of glucose [...] be rescued with acetate' did the authors mean to say that only acetate compared to glucose could recover the growth

defect due to inhibition of cholesterol metabolism? Please, clarifying in the text.

5. Lines 234-236

The authors should provide a more detailed description of the reasoning that led them to conclude that 3 molecules of sAEL057 coordinate a single Fe²⁺ ion.

6. Recent discoveries

Although the relevance of lipids and cholesterol metabolism during infection is well known, in recent years a growing body of experimental evidence has shown that Mtb is exposed to and capable of using carbon and energy sources other than lipids such as lactate and pyruvate (PMID: 31389636; PMID: 32034005; PMID: 28744015; PMID: 31201418). Therefore, the authors should take these recent findings into account when drafting their manuscript. An important revelation in recent years is the ability of Mtb to use a reverse methylcitrate cycle to assimilate carbon sources other than lipids and cholesterol. In light of this, certain clarifications are necessary to ensure that the results obtained are strictly related to a relationship between iron homeostasis and cholesterol metabolism:

♣ The 7H12 medium used for this study contains a pancreatic digest that represents a nutritional stimulus. Do the authors know the composition of this product? Are they sure that under such growth conditions the methylcitrate cycle goes in the canonical direction from propionyl-CoA to pyruvate?

♣ It is unclear how the use of the prpD'::GFP reporter can indicate that there is a blockage in propionyl-CoA production. Is it known whether prpD transcription is positively regulated by the presence of propionyl-CoA? Could a decrease in expression result in an accumulation of propionyl-CoA?

7. Lines 294-305

In this part of the discussion, attention is given to a result obtained in a previous work (reference 25), which is not further developed in this study. The link is not clear.

Reviewer #3 (Remarks to the Author):

Prevailing evidence points to the importance of fatty acids, such as cholesterol, as carbon sources for intracellular growth and survival of Mycobacterium tuberculosis (Mtb). Whilst there is growing appreciation for this and related phenomena, the regulatory circuitry of the appending pathways, together with how they can be exploited for tuberculosis drug discovery have remained largely elusive. The work of Theriault and colleagues is aimed at using a compound, that appears to inhibit mycobacteria through iron sequestration, as a tool for further understanding how limiting this essential nutrient affects bacterial metabolism. Using a high-throughput screening approach that assesses survival of intracellular (Mtb), combined with counter screens on fatty acids, the authors identify sAEL057 as an inhibitor of Mtb grown in macrophages and cholesterol, but not under nutrient rich conditions. The authors confirm the activity of this compound, demonstrating its carbon source-specific effect (high activity on cholesterol and glucose, the former able to be rescued by adding longer chain fatty acids to the media) and demonstrating bactericidal activity in broth and macrophages. To determine the mode of action of sAEL057, the authors conduct transcriptional profiling and demonstrate that treatment of Mtb with this agent induces global transcriptional changes that are distinct from other drugs/compounds. Amongst the comparators, the authors use another compound, mCLB073, which also demonstrated cholesterol-dependent antibacterial activity. However, despite this similarity, mCLB073 and sAEL057 displayed significantly different transcriptional changes in various aspects of metabolism. sAEL057 selectively dysregulates numerous genes involved in iron import or homeostasis. To explore the association with iron metabolism, the authors co-treated bacteria with sAEL057 and gallium, as the latter is a known inhibitor of iron-dependent enzymes and processes. Addition of gallium synergized with sAEL057-dependent bacterial inhibition on cholesterol but not nutrient rich media. This confirmed that the iron-associated effects of sAEL057 were only apparent in when bacteria were metabolising cholesterol. The authors proceed to demonstrate that

sAEL057 does not affect early steps in cholesterol catabolism and most likely works by inhibiting a step/s prior to metabolites entering the methylcitrate cycle. Finally, the authors demonstrate the knockdown of the transcriptional regulator, IdeR, phenocopies the effects of sAEL057. The work has been conducted well, with appropriate controls.

Major concerns:

1. In Figure 1D and E, higher doses of sAEL057 appear to give less killing. Do the authors have an explanation for this? At high concentrations, do select metabolic effects come into play? Can this be further explored with gene expression analysis?
2. The use of mass spec to confirm that sAEL057 binds iron is an important experiment to support the conclusions. It is strange that these experiments are done with pure compound and iron-containing compounds. Why did the authors not assess this in bacterial extracts isolated from organisms treated with the compound? Surely this would definitively demonstrate the intracellular mode of action. This should be done, on bacteria grown with cholesterol and on rich media.

In addition to individual responses to the reviewer comments below, we have also updated the figures and figure legends to be in compliance with the editorial policy checklist.

Reviewers' comments:

Reviewer #1 (Remarks to the Author):

The authors have identified a compound sAEL057 by a high-throughput screen, which demonstrates inhibitory activity on Mtb grown in conditions with cholesterol as the primary carbon source by limiting Mtb's access to iron. The methods are well-described and manuscript reads well.

While the authors have discussed the relevance of disrupting cholesterol metabolism and iron sequestration in controlling Mtb, the *in vivo* relevance of this study requires a more detailed explanation.

The data provide further justification of the validity of bacterial cholesterol metabolism as a drug target of value *in vivo*. This underpins data already published on the inhibition of cholesterol degradation through activation of the adenylate cyclase rv1625c (ref 25). This family of compounds is moving forwards as potential drugs against Mtb, and have been shown to be active in mice (ref 27).

It is unclear from the title and abstract what exactly the authors are proposing with respect to sAEL057. To establish the role of this compound in iron sequestration, a direct titration assay or a competition assay would be beneficial.

We believe that sAEL057 has limited potential as a drug because we have other, more effective compounds against cholesterol metabolism (V-059). However, as mentioned above we believe that data have value in verifying cholesterol metabolism as a drug target. We have modified both the title of the paper and the abstract to stress this point.

With respect to the titration assay, we believe that this is effectively what we have done using Gallium as a competitive inhibitor. We have clarified this in the revised manuscript.

Any specific reason for not using an iron chelator as a comparator in PCA to understand MOA?

Our first indication of potential mechanism came from the RNA-seq experiments, and therefore we didn't have any prior knowledge that sAEL057 activity was related to iron sequestration. The signature is an extremely clean one and had been detailed in depth previously, as referenced in the Dual RNA-seq study published previously (12).

A small section in discussion and a proposed model figure will be beneficial for the readers to understand the interrelationship proposed by the authors.

Thanks for the suggestion, a model figure (Fig. 6) has been added to the discussion section.

Line 136 mentions MOI of 1:1 for assessing drug activity in HMDMs while line 376 mentions at MOI 2:1. Kindly clarify.

Thank you for pointing this out, we have updated the text to include the correct MOI.

Reviewer #2 (Remarks to the Author):

“Iron limitation in *M. tuberculosis* has broad impact on bacterial metabolism revealing alternative routes to novel therapeutics” by Theriault M.E. et al. describes a study that identifying a new inhibitor of *Mycobacterium tuberculosis* replication in macrophage, and it reveals that this inhibitor is a iron chelator able to inhibit the use of cholesterol by the bacteria.

This study is of great importance because it adds to the limited knowledge concerning the link between metal homeostasis and carbon metabolism, an aspect of bacterial physiology that has only recently been emphasised (PMID: 34080971). The authors posing special attention to link between cholesterol degradation and iron homeostasis.

However, the work shows some limitations as follows:

1. Unfulfilled expectations

The title is full of expectation, as the work focuses only on cholesterol, leaving out the phenotype observed on glucose. It is unclear why the effect of sAEL057 on growth in 7H12 glucose was not further investigated by the authors. It would add value to the work by better supporting the "broad impact of iron limitation on bacterial metabolism". The experiments shown in Figures 3 and 5 could be completed on *Mtb* grown on 7H12 glucose.

Thank you for this suggestion. We completed these two requested experiments in 7H9 + glucose and they are shown in Supplemental Figures 5 and 8-9.

2. High throughput screening.

The minimum medium used for screening bacterial cultures is not specified in the materials and methods. In addition, a figure would make it clear how the authors identified the compound sAEL057. The authors could insert a figure like the one used in their previous work (figure 1 of reference 25).

We have extended the first section in M&M to include this information. sAEL057 was identified in a very similar manner to the cholesterol inhibitors discussed in reference 25, which we have indicated in the first section of the results.

3. 7H12 medium

It is not clear whether Rich Broth consists of 7H12 or 7H9 supplemented with OADC. And therefore, if Rich Broth consists of 7H9 supplemented with OADC, the experiments in figures 1, 3 and 5 should be repeated using as control condition 7H12 + OADC. 7H12 contains a pancreatic digest that represents a nutritional stimulus that cannot be underestimated.

Thank you for bringing this to our attention. Our lab uses the name "7H12" when discussing minimal media versus rich broth. However, all broth conditions were made using 7H9 base (7H12 base was not used for any experiment in this study). We have updated the text, figures/figure legends, and methods to reflect this and prevent confusion for readers.

4. Lines 116-118

In the sentence 'we noted [...] could not be rescued by addition of glucose [...] be rescued with acetate' did the authors mean to say that only acetate compared to glucose could recover the growth defect due to inhibition of cholesterol metabolism? Please, clarifying in the text.

5. Lines 234-236

The authors should provide a more detailed description of the reasoning that led them to conclude that 3 molecules of sAEI057 coordinate a single Fe²⁺ ion.

We have included more detailed information based on the LC/MS analysis. Interestingly, the trimer structure is actually quite common amongst Fe²⁺ chelators.

6. Recent discoveries

Although the relevance of lipids and cholesterol metabolism during infection is well known, in recent years a growing body of experimental evidence has shown that Mtb is exposed to and capable of using carbon and energy sources other than lipids such as lactate and pyruvate (PMID: 31389636; PMID: 32034005; PMID: 28744015; PMID: 31201418). Therefore, the authors should take these recent findings into account when drafting their manuscript. An important revelation in recent years is the ability of Mtb to use a reverse methylcitrate cycle to assimilate carbon sources other than lipids and cholesterol. In light of this, certain clarifications are necessary to ensure that the results obtained are strictly related to a relationship between iron homeostasis and cholesterol metabolism:

♣ The 7H12 medium used for this study contains a pancreatic digest that represents a nutritional stimulus. Do the authors know the composition of this product? Are they sure that under such growth conditions the methylcitrate cycle goes in the canonical direction from propionyl-CoA to pyruvate?

Addressed above, 7H12 medium, as understood by the reviewer, was not used in this study, and this has been clarified in the text.

♣ It is unclear how the use of the prpD':GFP reporter can indicate that there is a blockage in propionyl-CoA production. Is it known whether prpD transcription is positively regulated by the presence of propionyl-CoA? Could a decrease in expression result in an accumulation of propionyl-CoA?

Propionyl-CoA induces expression of prpD through positive regulation, which we have further clarified in the text and included additional references.

7. Lines 294-305

In this part of the discussion, attention is given to a result obtained in a previous work (reference 25), which is not further developed in this study. The link is not clear.

We have introduced this observation earlier in the manuscript (lines 120 – 125) where we believe it links more effectively with our existing knowledge on the metabolic wiring associated with cholesterol metabolism in Mtb.

Reviewer #3 (Remarks to the Author):

Prevailing evidence points to the importance of fatty acids, such as cholesterol, as carbon sources for intracellular growth and survival of *Mycobacterium tuberculosis* (Mtb). Whilst there is growing appreciation for this and related phenomena, the regulatory circuitry of the appending pathways, together with how they can be exploited for tuberculosis drug discovery have remained largely elusive. The work of Theriault and colleagues is aimed at using a compound, that appears to inhibit mycobacteria through iron sequestration, as a tool for further understanding how limiting this essential nutrient affects bacterial metabolism. Using a high-throughput screening approach that assesses survival of intracellular (Mtb), combined with counter screens on fatty acids, the authors identify sAEL057 as an inhibitor of Mtb grown in macrophages and cholesterol, but not under nutrient rich conditions. The authors confirm the activity of this compound, demonstrating its carbon source-specific effect (high activity on cholesterol and glucose, the former able to be rescued by adding longer chain fatty acids to the media) and demonstrating bactericidal activity in broth and macrophages. To determine the mode of action of sAEL057, the authors conduct transcriptional profiling and demonstrate that treatment of Mtb with this agent induces global transcriptional changes that are distinct from other drugs/compounds. Amongst the comparators, the authors use another compound, mCLB073, which also demonstrated cholesterol-dependent antibacterial activity. However, despite this similarity, mCLB073 and sAEL057 displayed significantly different transcriptional changes in various aspects of metabolism. sAEL057 selectively dysregulates numerous genes involved in iron import or homeostasis. To explore the association with iron metabolism, the authors co-treated bacteria with sAEL057 and gallium, as the latter is a known inhibitor of iron-dependent enzymes and processes. Addition of gallium synergized with sAEL057-dependent bacterial inhibition on cholesterol but not

nutrient rich media. This confirmed that the iron-associated effects of sAEL057 were only apparent when bacteria were metabolising cholesterol. The authors proceed to demonstrate that sAEL057 does not affect early steps in cholesterol catabolism and most likely works by inhibiting a step/s prior to metabolites entering the methylcitrate cycle. Finally, the authors demonstrate the knockdown of the transcriptional regulator, IdeR, phenocopies the effects of sAEL057. The work has been conducted well, with appropriate controls.

Major concerns:

1. In Figure 1D and E, higher doses of sAEL057 appear to give less killing. Do the authors have an explanation for this? At high concentrations, do select metabolic effects come into play? Can this be further explored with gene expression analysis?

We can't say with certainty what the concentration dependent impacts are on specific metabolic pathways due to the complexity of iron homeostasis and metabolomic programming in Mtb. In HMDMs there may be a trivial explanation, in that the highest concentration of sAEL057 appears to be impacting cell fitness.

2. The use of mass spec to confirm that sAEL057 binds iron is an important experiment to support the conclusions. It is strange that these experiments are done with pure compound and iron-containing compounds. Why did the authors not assess this in bacterial extracts isolated from organisms treated with the compound? Surely this would definitively demonstrate the intracellular mode of action. This should be done, on bacteria grown with cholesterol and on rich media.

We chose to perform mass spectrometry on pure sAEL057 and simple iron sources to show unequivocally that sAEL057 physically binds to iron. Due to the result of sAEL057 only binding to ferrous iron, we hypothesize that sAEL057 is binding to intracellular iron, based on knowledge of iron import in Mtb. Most iron inside the macrophage is in a ferric state and it is only available to Mtb in its ferrous form when it is released from mycobactin molecules in the bacterial cytosol. This suggests that sAEL057 binds to ferrous iron after it is released from the mycobactin (therefore intracellular), and now detailed in the model in Figure 6. We fully agree having mass spec results from Mtb extracts would be of value. However, to generate these extracts and prepare them for mass spec outside of a BSL3 poses a major safety concern for our core facilities. Additionally, we expect detection of a low concentration of sAEL057 inside a complex extract would be extremely challenging. Lastly, to our knowledge, and we did look extensively, this type of experiment has not been done on any other bacteria much less one as infectious and complex as Mtb.

REVIEWERS' COMMENTS:

Reviewer #1 (Remarks to the Author):

The authors have now answered all the queries raised and manuscript can be accepted for publication.

Reviewer #2 (Remarks to the Author):

The authors addressed all the issues presented. However, there are two points that should be further discussed.

1. The importance of lipid metabolism during infection is unquestionable, but a substantial body of evidence indicates that Mtb is exposed to and capable of using other substrates, other than lipids, during infection, and that appear to be equally important (PMID: 31389636; PMID: 32034005; PMID: 28744015; PMID: 31201418). Furthermore, a recent study found that host fatty acids are a direct source of carbon for the mycobacterial envelope without passing through β -oxidation, indeed their catabolism appear detrimental for mycobacteria replication (PMID: 33853942).

Therefore, I disagree with two statements the authors make in the text:

Lines 24-25

"...lipids serve as the primary carbon and energy source for Mtb in vivo and fulfil major roles in Mtb physiology and pathogenesis..."

Line 77

"...Mtb preferentially catabolizes lipid substrates¹⁶⁻²⁰ (cholesterol and fatty acids)..."

2. The citation of most of the figures is wrong, and there are some typing errors.

Reviewer #3 (Remarks to the Author):

The authors have addressed my concerns satisfactorily. Figure 6 is cut-off in the uploaded version, presumably this will be corrected in due course

REVIEWERS' COMMENTS:

Reviewer #1 (Remarks to the Author):

The authors have now answered all the queries raised and manuscript can be accepted for publication.

Reviewer #2 (Remarks to the Author):

The authors addressed all the issues presented. However, there are two points that should be further discussed.

1. The importance of lipid metabolism during infection is unquestionable, but a substantial body of evidence indicates that *Mtb* is exposed to and capable of using other substrates, other than lipids, during infection, and that appear to be equally important (PMID: 31389636; PMID: 32034005; PMID: 28744015; PMID: 31201418). Furthermore, a recent study found that host fatty acids are a direct source of carbon for the mycobacterial envelope without passing through β -oxidation, indeed their catabolism appear detrimental for mycobacteria replication (PMID: 33853942).

Thank you for this point, we have added the following sentence: “The complexity of *Mtb*'s metabolic program has been further revealed through recent findings demonstrating *Mtb*'s capacity to catabolize host-derived glycolytic intermediates such as pyruvate and lactate^{23,24}” and included references PMID: 31389636 and PMID: 28744015.

Therefore, I disagree with two statements the authors make in the text:

Lines 24-25

“...lipids serve as the primary carbon and energy source for *Mtb* in vivo and fulfil major roles in *Mtb* physiology and pathogenesis...”

This has been changed to “lipids are actively catabolized by *Mtb* in vivo and fulfill major roles in *Mtb* physiology and pathogenesis”

Line 77

“...*Mtb* preferentially catabolizes lipid substrates 16-20 (cholesterol and fatty acids)....”

This has been changed to “*Mtb* actively catabolizes”

2. The citation of most of the figures is wrong, and there are some typing errors.

Thank you for pointing this out, the numbering has been fixed and the manuscript read thoroughly for errors.

Reviewer #3 (Remarks to the Author):

The authors have addressed my concerns satisfactorily. Figure 6 is cut-off in the uploaded version, presumably this will be corrected in due course

Thank you, this has been fixed.